# Improving Perceptual Quality of Adversarial Images Using Perceptual Distance Minimization and Normalized Variance Weighting

**Berat Tuna Karli, Deniz Sen, Alptekin Temizel**

Graduate School of Informatics, Middle East Technical University
tuna.karli@metu.edu.tr, deniz.sen_01@metu.edu.tr, atemizel@metu.edu.tr

## Abstract

Neural networks are known to be vulnerable to adversarial examples, which are obtained by adding intentionally crafted perturbations to original images. However, these perturbations degrade their perceptual quality and make them more difficult to perceive by humans. In this paper, we propose two separate attack agnostic methods to increase the perceptual quality, measured in terms of perceptual distance metric LPIPS, while preserving the target fooling rate. The first method intensifies the perturbations in the high variance areas in the images. This method could be used in both white-box and black-box settings for any type of adversarial examples with only the computational cost of calculating the pixel based image variance. The second method aims to minimize the perturbations of already generated adversarial examples independent of the attack type. In this method, the distance between benign and adversarial examples are reduced until adversarial examples reach the decision boundaries of the true class. We show that these methods could also be used in conjunction to improve the perceptual quality of adversarial examples and demonstrate the quantitative improvements on CIFAR-10 and NIPS2017 Adversarial Learning Challenge datasets.

## Introduction

While Deep Neural Networks (DNNs) are being used in a variety of domains, there are several studies that show their vulnerabilities. An initial study, L-BFGS method (Szegedy et al. 2014), revealed that neural networks are not robust to adversarial attacks specifically produced to fool the networks. After the discovery of adversarial attacks, several different methods have been proposed such as Fast Gradient Sign Method (FGSM) (Goodfellow, Shlens, and Szegedy 2015), Projected Gradient Descent (PGD) (Madry et al. 2018), DeepFool (Moosavi-Dezfooli, Fawzi, and Frossard 2016), Jacobian Saliency Map Attack (JSMA) (Papernot et al. 2016), Spatially Transformed Adversarial Examples (stAdv) (Xiao et al. 2018) and Carlini&Wagner Attack (Carlini and Wagner 2017).

As adversarial examples can fool the networks, they can be used for the purpose of distinguishing humans from algorithms. While humans could still perceive the content of

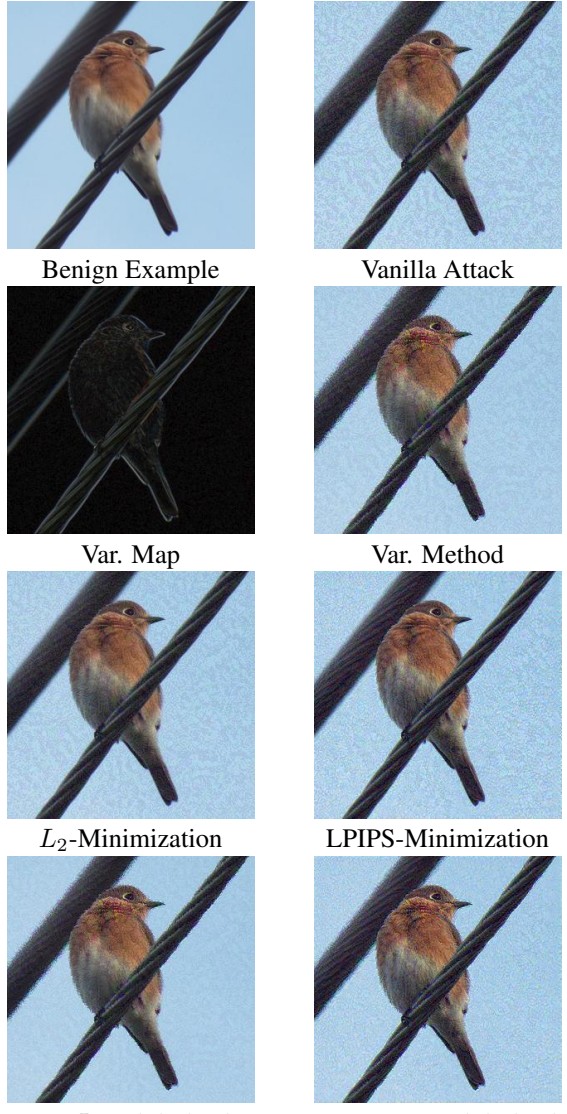

Benign Example | Vanilla Attack

Var. Map | Var. Method

$L_2$-Minimization | LPIPS-Minimization

Var. + $L_2$-Minimization | Var. + LPIPS-Minimization

Figure 1: FGSM ($\epsilon = 12/255$) results on NIPS2017 against ResNet50 with both proposed methods: variance weighting and minimization (shown separately for minimization with respect to $L_2$ and LPIPS) and combinations of them.

these images, algorithms would be deceived by the adversarial input. For such a system be effective, the perturbation that will be added to the image should be reasonably small and, while still misleading the algorithm, human vision should not be distracted from by the perturbation. Completely Automated Public Turing test to tell Computers and Humans Apart - CAPTCHA, is one of the most common examples where human users are distinguished from computer algorithms (Aksoy and Temizel 2019). The main motivation of this study is to improve successful adversarial attacks while reducing the perturbations that are distracting to humans. So, we propose two separate methods to improve the perceptual quality while keeping the attacks successful. The first method is based on intensifying the perturbation in high-variance zones and suppressing in low-variance zones using the variance map of input images for any type of attack. In effect, disguising the adversarial noise in high-variance areas and limiting the high-frequency noise added to low-variance areas where they would be more distracting. The second method is based on minimization of the perturbation until it reaches the boundary. While variance weighting is applied during the attack, minimization method could be considered as post processing after acquiring the adversarial example with any type of adversarial attack. As seen in Figure 1, localized and minimized perturbations improve the perceptual quality while keeping the fooling rate stable.

## Related Work

### Adversarial Attacks

L-BFGS is the initial method for generating adversarial examples using box-constrained optimization method (Szegedy et al. 2014). However, this method is computationally very costly. FGSM (Goodfellow, Shlens, and Szegedy 2015) is an efficient gradient-based attack algorithm, which computes the gradient only once, and adds perturbation in the gradient ascending direction of the loss function. Iterative Fast Gradient Sign Method (I-FGSM) (Kurakin, Goodfellow, and Bengio 2017) extends FGSM by iteratively attacking with a small step size and calculating the gradient at each step. C&W attack (Carlini and Wagner 2017) minimizes $L_2$ norm with an improved optimization method. DeepFool (Moosavi-Dezfooli, Fawzi, and Frossard 2016) efficiently computes the smallest perturbations according to closest decision boundary. Jacobian-based Saliency Map Attack (JSMA) (Papernot et al. 2016) generates sparse perturbations via generating saliency map and rank the contribution of each input variable to the adversarial objective. A perturbation is then selected from the saliency map at each iteration.

### Perceptual Metrics

All adversarial attack methods essentially aim to fool the network while minimizing the dissimilarity between benign and adversarial examples (i.e., minimizing the added perturbation). While the similarity metrics vary according to attack type, the most widely used distance metrics are $L_p$ norms ($p$ = 0, 1, 2, $\infty$). In particular, FGSM is an $L_\infty$, JSMA is an $L_0$, and C&W is an $L_2$ norm based attack. Even

though $L_p$ norms are very convenient and commonly used, several studies state that $L_p$ norms do not reflect the human perception accurately (Sharif, Bauer, and Reiter 2018; Jordan et al. 2019). Besides, there are some attack types such as (Jordan et al. 2019; Laidlaw and Feizi 2019; Laidlaw, Singla, and Feizi 2021; Aydin et al. 2021) for which $L_p$ norms are not fully suitable to evaluate the attack success. Thus these studies employ different and more recent perceptual metrics such as Learned Perceptual Image Patch Similarity (LPIPS) metric (Zhang et al. 2018) or Deep Image Structure and Texture Similarity (DISTS) index (Ding et al. 2020). Both of these methods use an additional neural network to measure the distance. LPIPS is calibrated with human perception and measures the Euclidean distance of deep representations. Likewise, DISTS optimizes human perception while using the combination of deep image structure and texture similarity.

### Variance Map on Adversarial Attacks

Human perception is affected more by perturbations in the low variance areas compared to high variance areas and this information is exploited in various image processing applications (Legge and Foley 1980; Lin, Dong, and Xue 2005; Liu et al. 2010). Regarding this fact, variance map has been used in previous studies (Luo et al. 2018; Croce and Hein 2019) to generate adversarial examples. In this work variance map is used to produce a variance based componentwise box constraints to generate sparse adversarial examples (Croce and Hein 2019). In another study, variance map is applied for the selection of high variance pixels (Luo et al. 2018). Using only $L_p$ norms for these variance based sparse attacks do not accurately reflect the perceptual quality (Luo et al. 2018), thus variance based sparse adversarial examples either use mean and median values of pixels or introduce a new distance metric that is more suited for the evaluation of their proposed attacks.

## Methodology

### Normalized Variance Weighting

In our study, we use variance map to intensify the perturbations in the high variance zones, instead of selecting high variance pixels or variance boundaries in an attack agnostic manner. We adopt the variance map method in (Croce and Hein 2019) to produce variance map of input images. In this method, standard deviation of both axes with 2 neighbour pixels and main pixel for each color channel are calculated ($\sigma_{ij}^{(x)}$ and $\sigma_{ij}^{(y)}$ for $x$ and $y$ axis respectively) and the square root of the minimum of standard deviation of axes is taken to obtain variance map $\sigma_{ij}$ (Equation 1). The variance map is then normalized to obtain normalized variance map $V_{i,j}$ (Equation 2).

$$\sigma_{ij} = \sqrt{\min\left\{\sigma_{ij}^{(x)}, \sigma_{ij}^{(y)}\right\}} \tag{1}$$

$$V_{i,j} = \frac{\sigma_{i,j}}{\sqrt{\sum_h^H \sum_w^W \sigma_{h,w}^2}} \tag{2}$$

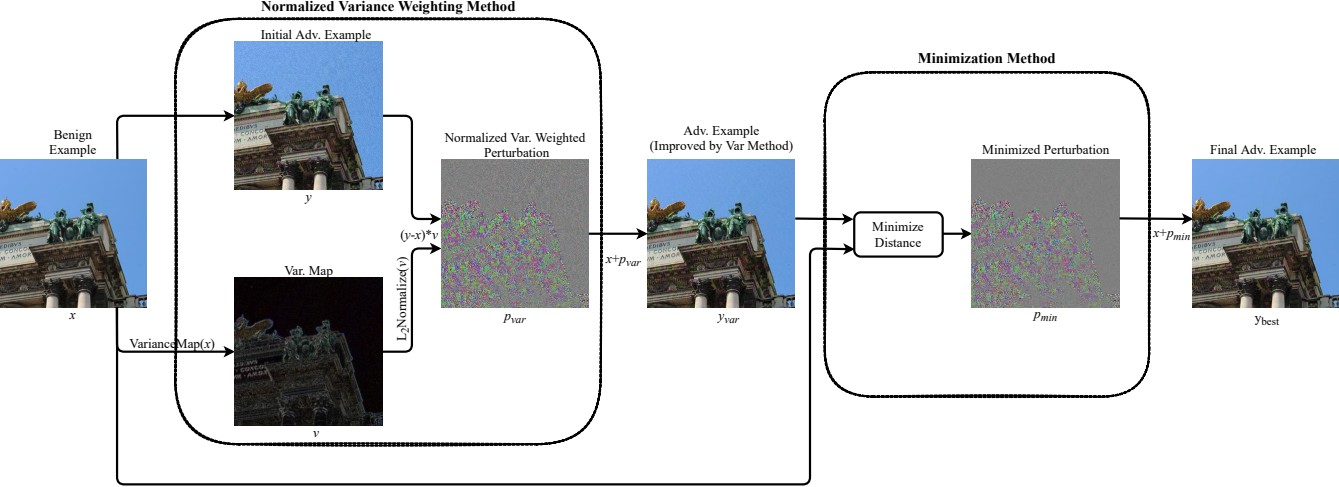

Figure 2: Visual representation of integration of proposed methods.

Since our method does not involve selecting pixels or generating variance box constraints, it does not require any additional threshold or coefficient variable. Normalizing and weighting procedures remove the need for an additional variable. As seen in Algorithm 1, the proposed method initially generates the variance map of input images for only once (Equation 1), then normalizes the variance map using $L_2$-norm (Equation 2) and applies variance map by weighting the perturbation with normalized variance map at each iteration (if adversarial attack is iterative). This method could be adapted for both white-box and black-box setting and does not require any optimization, or additional gradient-based steps. Therefore it does not bring any additional computational cost except the calculation of variance map for once.

## Minimization Method

The proposed minimization method is applied after generating the initial adversarial example and aims to reduce the distance between benign and adversarial examples using an optimizer. Optimizer minimizes the distance until adversarial examples reach the decision boundaries of true classes or maximum iteration number (Algorithm 2). We apply our minimization technique to minimize with regards to two different: $L_2$-norm and LPIPS (It has to be noted that some attacks are not fully suitable for $L_2$ distance metric (Aydin et al. 2021)). As LPIPS measures the perceptual distance using an additional neural network (i.e., VGG16 (Simonyan and Zisserman 2015)), it has higher processing time and higher number of parameters compared to $L_2$-norm minimization. In (Aksoy and Temizel 2019), the attack strength is iteratively adjusted to obtain the minimal perturbation needed in an attack agnostic manner after the generation of adversarial example, our proposed method improves this by directly optimizing the minimization of perturbation.

---

**Algorithm 1: Normalized Variance Weighting**

**Input**: $x$: original image, $Adv$: one iteration of adversarial attack
**Parameter**: $i_{max}$: maximum iteration of adversarial attack
**Output**: $y$: adversarial example

1: Let $i = 0$
2: $v = VarianceMap(x)$
3: $v = L_2Normalize(v)$
4: $y = x$
5: **while** $i < i_{max}$ **do**
6:     $y = Adv(y)$
7:     $p = (y - x) \times v$
8:     $y = x + p$
9:     $i = i + 1$
10: **end while**
11: **return** y

---

## Normalized Variance Weighting + Minimization

Normalized variance weighting method is applied during the adversarial attack while minimization method is applied after the generation of adversarial example. So both methods could be integrated and used in association for generation of adversarial examples. The complete pipeline integrating both methods is illustrated in Figure 2. We first generate adversarial examples, apply the variance weighting method and after the generation of variance weighted adversarial example, we apply minimization method as a post processing to obtain improved adversarial examples.

## Experiments

**Datasets.** We used CIFAR-10 and NIPS2017 Adversarial Learning Challenge datasets in the experiments. CIFAR-10 testset contains 10000 images with $32 \times 32$ resolution. We conducted our experiments on a subset of CIFAR-10 testset with 1000 images (100 random images from each category). NIPS2017 dataset is a subset of Imagenet dataset and

Table 1: FGSM results on CIFAR10 dataset against ResNet50 with and without variance weighting (shown as Var.) and minimization method (shown as Minim.) using LPIPS and $L_2$. Results are reported in both LPIPS ($\times 10^2$) and $L_2$ metrics.

| Var. | Minim. | 30% | | 40% | | 50% | | 60% | |
|------|--------|-------|-------|-------|-------|-------|-------|-------|-------|
| | | LPIPS | $L_2$ | LPIPS | $L_2$ | LPIPS | $L_2$ | LPIPS | $L_2$ |
| - | - | 0.19 | 0.13 | 0.93 | 0.28 | 3.19 | 0.58 | 7.42 | 1.14 |
| - | LPIPS | 0.07 | 0.13 | 0.59 | 0.27 | 2.58 | 0.57 | 6.46 | 1.13 |
| - | $L_2$ | 0.10 | **0.09** | 0.75 | **0.22** | 2.93 | 0.52 | 7.06 | **1.09** |
| + | - | 0.16 | 0.14 | 0.54 | 0.26 | 2.36 | 0.54 | 6.85 | 1.18 |
| + | LPIPS | **0.06** | 0.13 | **0.29** | 0.25 | **1.82** | 0.54 | **5.96** | 1.17 |
| + | $L_2$ | 0.09 | 0.10 | 0.41 | **0.22** | 2.14 | **0.50** | 6.54 | 1.13 |

Table 2: I-FGSM results on CIFAR10 dataset against ResNet50 with and without variance weighting (shown as Var.) and minimization method (shown as Minim.) using LPIPS and $L_2$. Results are reported in both LPIPS ($\times 10^2$) and $L_2$ metrics.

| Var. | Minim. | 60% | | 70% | | 80% | | 90% | |
|------|--------|-------|-------|-------|-------|-------|-------|-------|-------|
| | | LPIPS | $L_2$ | LPIPS | $L_2$ | LPIPS | $L_2$ | LPIPS | $L_2$ |
| - | - | 0.62 | 0.26 | 0.97 | 0.33 | 1.52 | 0.41 | 2.71 | 0.57 |
| - | LPIPS | 0.28 | 0.25 | 0.50 | 0.32 | 0.86 | 0.40 | 1.74 | 0.56 |
| - | $L_2$ | 0.43 | **0.21** | 0.73 | **0.28** | 1.22 | **0.37** | 2.32 | **0.52** |
| + | - | 0.57 | 0.27 | 0.87 | 0.34 | 1.37 | 0.43 | 2.51 | 0.60 |
| + | LPIPS | **0.26** | 0.26 | **0.43** | 0.33 | **0.76** | 0.42 | **1.58** | 0.59 |
| + | $L_2$ | 0.41 | 0.23 | 0.66 | 0.30 | 1.11 | 0.39 | 2.16 | 0.56 |

---

**Algorithm 2: Minimization Method**

---

**Input**: $x$: original image, $y$: adversarial example
**Parameter**: $lr$: learning rate, $i_{max}$: maximum iteration
**Output**: $y_{best}$: improved adversarial example

1: Let $i = 0$
2: $y_{best} = y$
3: $y_{opt} = y$
4: **while** $i < i_{max}$ **do**
5:    **if** $class_{y_{opt}} == class_x$ **then**
6:       **return** $y_{best}$
7:    **else**
8:       $y_{best} = y_{opt}$
9:    **end if**
10:   $y_{opt} = MinimizeDIST(y_{opt}, x, lr)$
11: **end while**
12: **return** $y_{best}$

---

contains 1000 images (one images from each category) with $299 \times 299$ resolution.

**Attack Types.** We have tested the proposed methods using 3 different untargeted attack types: a single step gradient based attack (FGSM) (Goodfellow, Shlens, and Szegedy 2015), an iterative gradient based attack (I-FGSM) (Kurakin, Goodfellow, and Bengio 2017) and an optimization based attack (C&W) (Carlini and Wagner 2017) on CIFAR10 and NIPS2017 datasets. We have used ResNet50 (He et al. 2016) and Inception-V3 (Szegedy et al. 2016) for NIPS2017 dataset; only ResNet50 (He et al. 2016) for CIFAR10 dataset. We have used CleverHans (Papernot et al. 2018) implementation for default attacks and we integrated the proposed methods into these attacks.

## Experimental Settings for Normalized Variance Weighting

For variance map, we used 2 neighbour pixels and main pixel for every color channel in the generation of variance map similar to (Croce and Hein 2019). We observed that using variance weighting method considerably decreases the fooling rate when the attack strength is fixed. Thus, to compare on a fair ground, we fixed the fooling rate and let the $\epsilon$ (for FGSM and I-FGSM attacks) or initial cost (for C&W attack) vary. This allowed reaching the target fooling rate within a $\pm 0.5\%$ error tolerance. We targeted 4 different fooling rates for FGSM: 30%, 40%, 50%, 60% and I-FGSM: 60%, 70%, 80%, 90% on both datasets. We used a single fooling rate for C&W attack on each dataset: 95% on CIFAR10 and 100% on NIPS2017 (for both ResNet50 and Inception-V3), since there is $L_2$-normalization after producing variance map, measuring $L_p$ norms would be misleading for variance weighting method. Therefore, we mainly used LPIPS perceptual distance metric, which is calibrated with human vision, for its evaluation.

## Experimental Settings for Minimization Method

For the experimental settings of the proposed minimization method, we used Adam (Kingma and Ba 2015) as the optimizer and set the maximum iteration number as 10. We set the learning rate as 0.0001 for CIFAR10 dataset for both minimization methods. We set learning rate as 0.0001 for $L_2$-Minimization and 0.00001 for LPIPS-minimization on NIPS2017 (for both ResNet50 and Inception-V3), since they were not converging with the same learning rate.

Table 3: FGSM results on NIPS2017 dataset against ResNet50 and Inception-V3 with and without variance weighting (shown as Var.) and minimization method (shown as Minim.) using LPIPS and $L_2$. Results are reported in both LPIPS ($\times 10^2$) and $L_2$ metrics.

| Var. | Minim. | 30% | | | | 40% | | | | 50% | | | | 60% | | | |
|---|---|---|---|---|---|---|---|---|---|---|---|---|---|---|---|---|---|
| | | ResNet50 | | Inc-V3 | | ResNet50 | | Inc-V3 | | ResNet50 | | Inc-V3 | | ResNet50 | | Inc-V3 | |
| | | LPIPS | $L_2$ | LPIPS | $L_2$ | LPIPS | $L_2$ | LPIPS | $L_2$ | LPIPS | $L_2$ | LPIPS | $L_2$ | LPIPS | $L_2$ | LPIPS | $L_2$ |
| - | - | 0.07 | 0.22 | 0.12 | 0.30 | 0.14 | 0.32 | 0.26 | 0.43 | 0.26 | 0.42 | 0.54 | 0.63 | 0.44 | 0.55 | 1.10 | 0.94 |
| - | LPIPS | 0.02 | 0.20 | 0.02 | 0.27 | 0.03 | 0.28 | 0.04 | 0.39 | 0.05 | 0.38 | 0.10 | 0.57 | 0.08 | 0.50 | 0.26 | 0.87 |
| - | $L_2$ | 0.05 | 0.16 | 0,07 | 0,20 | 0.10 | 0.21 | 0,11 | **0,24** | 0.13 | **0.25** | 0,22 | **0,37** | 0.23 | **0.35** | 0,30 | **0,39** |
| + | - | 0.04 | 0.26 | 0,06 | 0,35 | 0.08 | 0.37 | 0,13 | 0,49 | 0.14 | 0.49 | 0,25 | 0,71 | 0.22 | 0.62 | 0,52 | 1,04 |
| + | LPIPS | **0.01** | 0.23 | **0,01** | 0,32 | **0.02** | 0.34 | **0,02** | 0,45 | **0.03** | 0.46 | **0,05** | 0,66 | **0.05** | 0.58 | **0,13** | 0,99 |
| + | $L_2$ | 0.02 | **0.15** | 0,03 | **0,17** | 0.04 | **0.20** | 0,05 | 0,26 | 0.08 | 0.29 | 0,09 | 0,38 | 0.12 | 0.39 | 0,15 | 0,56 |

Table 4: I-FGSM Results on NIPS2017 dataset against ResNet50 and Inception-V3 with and without variance weighting (shown as Var.) and minimization method (shown as Minim.) using LPIPS and $L_2$. Results are reported in both LPIPS ($\times 10^2$) and $L_2$ metrics.

| Var. | Minim. | 60% | | | | 70% | | | | 80% | | | | 90% | | | |
|---|---|---|---|---|---|---|---|---|---|---|---|---|---|---|---|---|---|
| | | ResNet50 | | Inc-V3 | | ResNet50 | | Inc-V3 | | ResNet50 | | Inc-V3 | | ResNet50 | | Inc-V3 | |
| | | LPIPS | $L_2$ | LPIPS | $L_2$ | LPIPS | $L_2$ | LPIPS | $L_2$ | LPIPS | $L_2$ | LPIPS | $L_2$ | LPIPS | $L_2$ | LPIPS | $L_2$ |
| - | - | 0.11 | 0.32 | 0.23 | 0.43 | 0.16 | 0.38 | 0.37 | 0.56 | 0.23 | 0.46 | 0.63 | 0.75 | 0.40 | 0.63 | 1.19 | 1.08 |
| - | LPIPS | 0.02 | 0.28 | 0.04 | 0.39 | 0.03 | 0.34 | 0.07 | 0.51 | 0.04 | 0.42 | 0.12 | 0.69 | 0.07 | 0.58 | 0.28 | 1.01 |
| - | $L_2$ | 0.07 | **0.22** | 0.14 | **0.29** | 0.12 | **0.28** | 0.21 | **0.36** | 0.18 | **0.36** | 0.25 | **0.44** | 0.25 | **0.45** | 0.32 | **0.55** |
| + | - | 0.09 | 0.41 | 0.13 | 0.52 | 0.12 | 0.49 | 0.21 | 0.67 | 0.18 | 0.61 | 0.37 | 0.90 | 0.31 | 0.81 | 0.71 | 1.29 |
| + | LPIPS | **0.02** | 0.38 | **0.03** | 0.47 | **0.03** | 0.45 | **0.04** | 0.62 | **0.04** | 0.56 | **0.08** | 0.84 | **0.06** | 0.76 | **0.17** | 1.23 |
| + | $L_2$ | 0.06 | 0.28 | 0.08 | 0.35 | 0.09 | 0.36 | 0.13 | 0.45 | 0.13 | 0.47 | 0.15 | 0.59 | 0.18 | 0.61 | 0.20 | 0.78 |

Table 5: CW2 Results on CIFAR10 dataset against ResNet50 with and without variance weighting (shown as Var.) and minimization method (shown as Minim.) using LPIPS and $L_2$. Results are reported in both LPIPS ($\times 10^2$) and $L_2$ metrics.

| Var. | Minim. | LPIPS | $L_2$ |
|---|---|---|---|
| - | - | 0.25 | 0.27 |
| - | LPIPS | 0.15 | 0.27 |
| - | $L_2$ | 0.25 | 0.28 |
| + | - | 0.19 | 0.27 |
| + | LPIPS | **0.12** | **0.26** |
| + | $L_2$ | 0.19 | 0.28 |

Table 6: CW2 Results on NIPS2017 dataset against ResNet50 and Inception-V3 with and without variance weighting (shown as Var.) and minimization method (shown as Minim.) using LPIPS and $L_2$. Results are reported in both LPIPS ($\times 10^2$) and $L_2$ metrics.

| Var. | Minim. | ResNet50 | | Inc-V3 | |
|---|---|---|---|---|---|
| | | LPIPS | $L_2$ | LPIPS | $L_2$ |
| - | - | 0.25 | 0.27 | 0.33 | 0.38 |
| - | LPIPS | 0.15 | 0.27 | **0.17** | **0.37** |
| - | $L_2$ | 0.25 | 0.28 | 0.32 | 0.38 |
| + | - | 0.19 | 0.27 | 0.33 | 0.45 |
| + | LPIPS | **0.12** | **0.26** | 0.19 | 0.45 |
| + | $L_2$ | 0.19 | 0.28 | 0.32 | 0.46 |

## Results

**Normalized Variance Weighting.** The effect of variance weighting method on FGSM attack can be observed in Table 1 and Table 3 for CIFAR10 (using ResNet50) and NIPS2017 (using ResNet50 and Inception-V3) datasets respectively. The method reduces the LPIPS distances considerably in all cases, i.e. without minimization and when used together with minimization with respect to both $L_2$ and LPIPS. The corresponding results in Table 2 and Table 4 for I-FGSM and Table 5 and Table 6 for C&W attack confirm that these findings are pertinent to these attacks as well and variance weighting is effective in reducing the LPIPS distance for all attack types in question.

**Minimization Methods.** LPIPS-Minimization method applied on FGSM attack decreases the LPIPS distances considerably when used individually as well as when it is combined together with variance weighting for both CI-FAR10 (using ResNet50) and NIPS2017 (using ResNet50 and Inception-V3) datasets (Table 1 and Table 3). The corresponding results in Table 2 and Table 4 for I-FGSM and Table 5 and Table 6 for C&W attack confirm that these findings are pertinent to these attacks as well and LPIPS is effective in reducing the LPIPS distance for all attack types in question.

In addition to these results, LPIPS-Minimization method also improves $L_2$ distance considerably. Though, as expected, $L_2$-Minimization method results in the best $L_2$ distance improvements for FGSM and I-FGSM on both CI-FAR10 (using ResNet50) and NIPS2017 (using ResNet50 and Inception-V3) datasets. Considering C&W is already optimizing L2 distance, the improvement is relatively limited for C&W attack on both datasets.

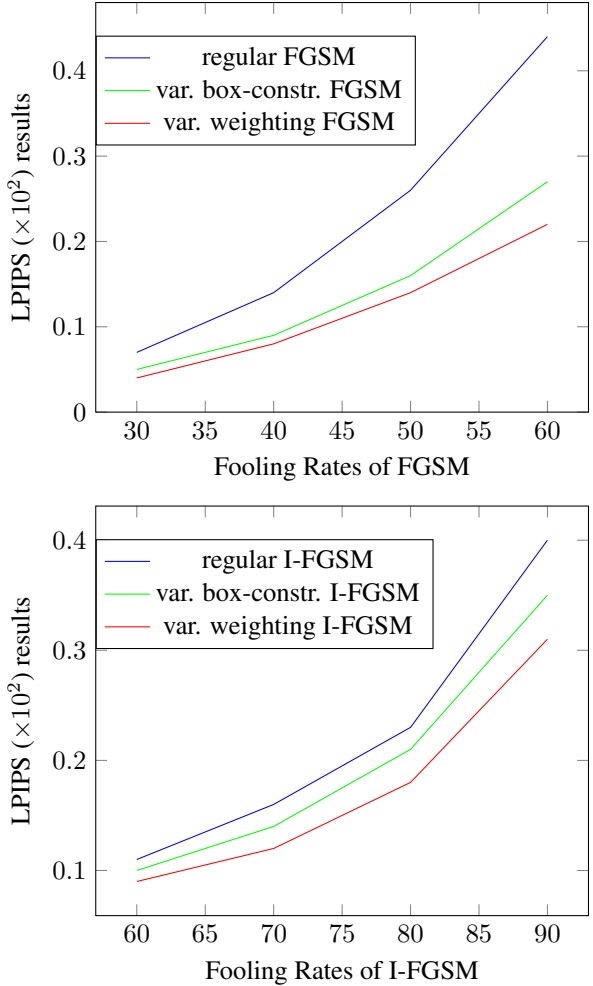

Figure 3: LPIPS results of FGSM and I-FGSM attacks on NIPS2017 dataset with regular method, normalized variance weighting method and variance-based box-constrained method using ResNet50.

**Variance + Minimization Methods.** The best results (i.e. lowest LPIPS distances) are obtained when we combine variance weighting method with LPIPS-Minimization and the results show that there is considerable improvement for FGSM, I-FGSM and C&W attack types on both CIFAR10 (using ResNet50) and NIPS2017 (using both ResNet50 and Inception-V3).

## Discussion

From the experiments, it is seen that both variance weighting and minimization methods individually improve the perceptual quality and the best LPIPS results are obtained when they are integrated. However, this improvement is relatively limited for attacks that are inherently able to produce adversarial examples with quantitatively lower perceptual distances, such as C&W. Results in Tables 1 to 6 show that NIPS2017 dataset results have smaller perceptual dis-

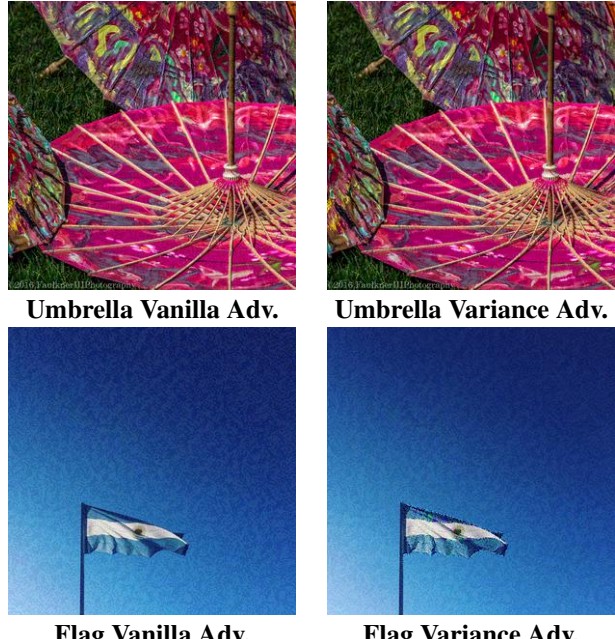

| **Umbrella Vanilla Adv.** | **Umbrella Variance Adv.** |
| **Flag Vanilla Adv.** | **Flag Variance Adv.** |

Figure 4: Ineffective NIPS2017 samples against normalized variance weighting Method.

tances yet improvement percentages are higher compared to CIFAR10 dataset (i.e., LPIPS distance is reduced by 10% and 25% for CIFAR10 (40% fooling rate) and NIPS2017 (70% fooling rate) datasets respectively for I-FGSM against ResNet50).

We have also investigated variance-based box-constrained method (Croce and Hein 2019) for our attack type agnostic white-box setting as an alternative to variance weighting. While variance-based box-constrained adversarial examples are improved in terms of perceptual quality as well, it is seen in Figure 3 that variance weighting method has lower LPIPS perceptual distances compared to variance-based box-constrained method at all fooling rates compared to FGSM and I-FGSM attacks on NIPS2017. Variance-based box-constrained method also requires an additional coefficient parameter for each adversarial attack type, dataset and network. In Figure 3, epsilon value is used as the additional coefficient parameter of variance-based box-constrained method to select adaptive coefficient. Even when parameters for threshold levels are optimized, in most instances, we have observed that variance weighted perturbations have better perceptual quality, which makes variance weighting a better choice.

We have conducted our experiments based on $L_2$ and LPIPS distance metrics. Since traditional $L_p$ norm based measures are based on pixel based differences, they are not effective indicators for perceptual quality, hence, we consider use LPIPS perceptual distance as the primary metric in our evaluation. With regards to the proposed minimization methods, LPIPS-Minimization can be used in conjunction with any type of attack, while $L_2$-Minimization is less effective in some attack types such as the ones based on shift-

ing of pixels (e.g., (Aydin et al. 2021)). Nevertheless, we have measured the distances of $L_2$ and LPIPS minimization methods with both $L_2$ and LPIPS metrics. It is seen that both distance metrics usually decrease with any of minimization method. Though, as expected, the measured metric benefits more when it is the same metric as the one used in minimization (e.g., LPIPS-minimization method minimizes LPIPS distance proportionally more compared to $L_2$ distance).

Our empirical observations show that the variance weighting method significantly improves perceptual quality of images which have low variance backgrounds (e.g., sky, wall or sea) as it could be seen in Figure 1. However it is less effective for images with dominantly high variance zones (e.g., umbrella image in Figure 4) and for images having dominantly low variance zones (e.g., flag image in Figure 4). In the flag image, the variance of the background is highly low and high variance region is very narrow, hence variance weighting method could not improve the flag image adequately.

## Conclusion

We have proposed two separate attack agnostic techniques to improve perceptual quality of adversarial examples while preserving the fooling rate. We have shown that applying our variance weighting improves the perceptual quality of different types of adversarial attacks without any significant computational cost in white-box setting. We have also shown that perturbations produced by different types of adversarial attacks could be minimized while preserving the fooling rate. Integration of the variance weighting and minimization generates adversarial examples with the best perceptual quality measured by LPIPS. Other attack agnostic improvements (e.g., generating adversarial attacks on YUV color space (Aksoy and Temizel 2019) could be combined with these two proposed methods to enhance perceptual quality further in the future.

## Acknowledgments

This work has been funded by The Scientific and Technological Research Council of Turkey, ARDEB 1001 Research Projects Programme project no: 120E093

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
