# OpenReview forum: "Improving Perceptual Quality of Adversarial Images Using Perceptual Distance Minimization and Normalized Variance Weighting"
_AAAI.org/2022/Workshop/AdvML — AAAI-22 AdvML Workshop LongPaper_

### Official Review · Reviewer_nVja · 2021-11-27
**Simple and generic methods**

**Rating:** 6
**Confidence:** 4

**Review:**

This paper mainly proposed two attack-agnostic methods with normalized variance weighting and perceptual distance minimization to enhance the perceptual quality of adversarial examples. The proposed algorithms are simple and can be combined with other methods in practice. Experiments are carried out in CIFAR-10 and NIPS2017 Adversarial Learning Challenge datasets, showing the combination of two methods indeed decrease the perceptual distance.

Strengths:
* The paper is well-organized.
* The proposed methods are simple but attack-agnostic, which increase the perceptual quality while preserve the fooling rate.

Weaknesses:
* The contributions are not adequate. The decrease of perceptual distance by minimizing the distance is obvious and inevitable, but why and how it can be helpful is not clear enough.  It seems that it doesn't work well for images with dominantly high or low variance zones, then the work seems not useful since the poor perceptual quality of adversarial examples often occurs in these cases.
* More analysis on whether the proposed methods may bring worse efficiency or transferability should be given. Also, the experimental of results comparing the normalized variance weighting and the variance-based box-constraints should be given instead of a casual remark.
* Some statements are confusing. Why the statement that ''our normalized variance weighting method is not suitable to measure for traditional Lp norms''  holds? More explanations should be given.
* There are some typos. For instance, in the algorithm2, I think the condition $class_{y_{opt}}\ne class_{x}$ is a mistake.

---

### Official Review · Reviewer_uSVu · 2021-11-30
**Review of Paper11**

**Rating:** 6
**Confidence:** 4

**Review:**

**Summary Of The Paper:**

This work studies the problem of the perceptual quality of adversarial images. The authors propose two separate attack agnostic methods to increase the perceptual quality. Experiments are conducted to demonstrate the efficacy of the proposed methods.

**Main Review:**

Strength:
* Interesting topic.
* Somehow intuitive.

Weakness:
* The paper lacks some theoretical analysis, for example, how to define the perceptual quality. The authors claim that the proposed methods can generate adversarial examples with the best perceptual quality. However, little quantitative evaluation is performed to assess the quality of the images. This type of anecdotal evidence is entirely unacceptable in academic work.
* The practical significance of the proposed methods is not clear, and it may be unnecessary in adversarial attacks.

---

### Decision · Program_Chairs · 2021-12-01

**Decision:**

Accept (Long Paper)

**Comment:**

The reviewers give positive ratings on this paper, although there are some concerns. Please address them in the camera-ready version.